# Sampling from Probabilistic Submodular Models

**Alkis Gotovos**
ETH Zurich
alkisg@inf.ethz.ch

**S. Hamed Hassani**
ETH Zurich
hamed@inf.ethz.ch

**Andreas Krause**
ETH Zurich
krausea@ethz.ch

## Abstract

Submodular and supermodular functions have found wide applicability in machine learning, capturing notions such as diversity and regularity, respectively. These notions have deep consequences for optimization, and the problem of (approximately) optimizing submodular functions has received much attention. However, beyond optimization, these notions allow specifying expressive probabilistic models that can be used to quantify predictive uncertainty via marginal inference. Prominent, well-studied special cases include Ising models and determinantal point processes, but the general class of log-submodular and log-supermodular models is much richer and little studied. In this paper, we investigate the use of Markov chain Monte Carlo sampling to perform approximate inference in general log-submodular and log-supermodular models. In particular, we consider a simple Gibbs sampling procedure, and establish two sufficient conditions, the first guaranteeing polynomial-time, and the second fast ($\mathcal{O}(n \log n)$) mixing. We also evaluate the efficiency of the Gibbs sampler on three examples of such models, and compare against a recently proposed variational approach.

## 1 Introduction

Modeling notions such as coverage, representativeness, or diversity is an important challenge in many machine learning problems. These notions are well captured by submodular set functions. Analogously, supermodular functions capture notions of smoothness, regularity, or cooperation. As a result, submodularity and supermodularity, akin to concavity and convexity, have found numerous applications in machine learning. The majority of previous work has focused on optimizing such functions, including the development and analysis of algorithms for minimization [10] and maximization [9, 26], as well as the investigation of practical applications, such as sensor placement [21], active learning [12], influence maximization [19], and document summarization [25].

Beyond optimization, though, it is of interest to consider probabilistic models defined via submodular functions, that is, distributions over finite sets (or, equivalently, binary random vectors) defined as $p(S) \propto \exp(\beta F(S))$, where $F : 2^V \to \mathbb{R}$ is a submodular or supermodular function (equivalently, either $F$ or $-F$ is submodular), and $\beta \geq 0$ is a scaling parameter. Finding most likely sets in such models captures classical submodular optimization. However, going beyond point estimates, that is, performing general probabilistic (e.g., marginal) inference in them, allows us to quantify uncertainty given some observations, as well as learn such models from data. Only few special cases belonging to this class of models have been extensively studied in the past; most notably, Ising models [20], which are log-supermodular in the usual case of attractive (ferromagnetic) potentials, or log-submodular under repulsive (anti-ferromagnetic) potentials, and determinantal point processes [23], which are log-submodular.

Recently, Djolonga and Krause [6] considered a more general treatment of such models, and proposed a variational approach for performing approximate probabilistic inference for them. It is natural to ask to what degree the usual alternative to variational methods, namely Monte Carlo sampling, is applicable to these models, and how it performs in comparison. To this end, in this paper

we consider a simple Markov chain Monte Carlo (MCMC) algorithm on log-submodular and log-supermodular models, and provide a first analysis of its performance. We present two theoretical conditions that respectively guarantee polynomial-time and fast ($\mathcal{O}(n \log n)$) mixing in such models, and experimentally compare against the variational approximations on three examples.

## 2 Problem Setup

We start by considering set functions $F : 2^V \to \mathbb{R}$, where $V$ is a finite ground set of size $|V| = n$. Without loss of generality, if not otherwise stated, we will hereafter assume that $V = [n] := \{1, 2, \ldots, n\}$. The marginal gain obtained by adding element $v \in V$ to set $S \subseteq V$ is defined as $F(v|S) := F(S \cup \{v\}) - F(S)$. Intuitively, submodularity expresses a notion of diminishing returns; that is, adding an element to a larger set provides less benefit than adding it to a smaller one. More formally, $F$ is submodular if, for any $S \subseteq T \subseteq V$, and any $v \in V \setminus T$, it holds that $F(v|T) \leq F(v|S)$. Supermodularity is defined analogously by reversing the sign of this inequality. In particular, if a function $F$ is submodular, then the function $-F$ is supermodular. If a function $m$ is both submodular and supermodular, then it is called modular, and may be written in the form $m(S) = c + \sum_{v \in S} m_v$, where $c \in \mathbb{R}$, and $m_v \in \mathbb{R}$, for all $v \in V$.

Our main focus in this paper are distributions over the powerset of $V$ of the form

$$p(S) = \frac{\exp(\beta F(S))}{Z}, \tag{1}$$

for all $S \subseteq V$, where $F$ is submodular or supermodular. The scaling parameter $\beta$ is referred to as inverse temperature, and distributions of the above form are called log-submodular or log-supermodular respectively. The constant denominator $Z := Z(\beta) := \sum_{S \subseteq V} \exp(\beta F(S))$ serves the purpose of normalizing the distribution and is called the partition function of $p$. An alternative and equivalent way of defining distributions of the above form is via binary random vectors $X \in \{0, 1\}^n$. If we define $V(X) := \{v \in V \mid X_v = 1\}$, it is easy to see that the distribution $p_X(X) \propto \exp(\beta F(V(X)))$ over binary vectors is isomorphic to the distribution over sets of (1). With a slight abuse of notation, we will use $F(X)$ to denote $F(V(X))$, and use $p$ to refer to both distributions.

**Example models** The (ferromagnetic) Ising model is an example of a log-supermodular model. In its simplest form, it is defined through an undirected graph $(V, E)$, and a set of pairwise potentials $\sigma_{v,w}(S) := 4(\mathbb{1}_{\{v \in S\}} - 0.5)(\mathbb{1}_{\{w \in S\}} - 0.5)$. Its distribution has the form $p(S) \propto \exp(\beta \sum_{\{v,w\} \in E} \sigma_{v,w}(S))$, and is log-supermodular, because $F(S) = \sum_{\{v,w\} \in E} \sigma_{v,w}(S)$ is supermodular. (Each $\sigma_{v,w}$ is supermodular, and supermodular functions are closed under addition.)

Determinantal point processes (DPPs) are examples of log-submodular models. A DPP is defined via a positive semidefinite matrix $K \in \mathbb{R}^{n \times n}$, and has a distribution of the form $p(S) \propto \det(K_S)$, where $K_S$ denotes the square submatrix indexed by $S$. Since $F(S) = \ln \det(K_S)$ is a submodular function, $p$ is log-submodular. Another example of log-submodular models are those defined through facility location functions, which have the form $F(S) = \sum_{\ell \in [L]} \max_{v \in S} w_{v,\ell}$, where $w_{v,\ell} \geq 0$, and are submodular. If $w_{v,\ell} \in \{0, 1\}$, then $F$ represents a set cover function.

Note that, both the facility location model and the Ising model use decomposable functions, that is, functions that can be written as a sum of simpler submodular (resp. supermodular) functions $F_\ell$:

$$F(S) = \sum_{\ell \in [L]} F_\ell(S). \tag{2}$$

**Marginal inference** Our goal is to perform marginal inference for the distributions described above. Concretely, for some fixed $A \subseteq B \subseteq V$, we would like to compute the probability of sets $S$ that contain all elements of $A$, but no elements outside of $B$, that is, $p(A \subseteq S \subseteq B)$. More generally, we are interested in computing conditional probabilities of the form $p(A \subseteq S \subseteq B \mid C \subseteq S \subseteq D)$. This computation can be reduced to computing unconditional marginals as follows. For any $C \subseteq V$, define the contraction of $F$ on $C$, $F_C : 2^{V \setminus C} \to \mathbb{R}$, by $F_C(S) = F(S \cup C) - F(S)$, for all $S \subseteq V \setminus C$. Also, for any $D \subseteq V$, define the restriction of $F$ to $D$, $F^D : 2^D \to \mathbb{R}$, by $F^D(S) = F(S)$, for all $S \subseteq D$. If $F$ is submodular, then its contractions and restrictions are also submodular, and, thus, $(F_C)^D$ is submodular. Finally, it is easy to see that $p(S \mid C \subseteq S \subseteq D) \propto \exp(\beta (F_C)^D(S))$. In

---
**Algorithm 1** Gibbs sampler
---
**Input:** Ground set $V$, distribution $p(S) \propto \exp(\beta F(S))$
 1: $X_0 \leftarrow$ random subset of $V$
 2: **for** $t = 0$ **to** $N_{\text{iter}}$ **do**
 3:     $v \leftarrow \text{Unif}(V)$
 4:     $\Delta_F(v|X_t) \leftarrow F(X_t \cup \{v\}) - F(X_t \setminus \{v\})$
 5:     $p_{\text{add}} \leftarrow \exp(\beta \Delta_F(v|X_t))/(1 + \exp(\beta \Delta_F(v|X_t)))$
 6:     $z \leftarrow \text{Unif}([0, 1])$
 7:     **if** $z \leq p_{\text{add}}$ **then** $X_{t+1} \leftarrow X_t \cup \{v\}$ **else** $X_{t+1} \leftarrow X_t \setminus \{v\}$
 8: **end for**
---

our experiments, we consider computing marginals of the form $p(v \in S \mid C \subseteq S \subseteq D)$, for some $v \in V$, which correspond to $A = \{v\}$, and $B = V$.

## 3 Sampling and Mixing Times

Performing exact inference in models defined by (1) boils down to computing the partition function $Z$. Unfortunately, this is generally a #P-hard problem, which was shown to be the case even for Ising models by Jerrum and Sinclair [17]. However, they also proposed a sampling-based FPRAS for a class of ferromagnetic models, which gives us hope that it may be possible to efficiently perform approximate inference in more general models under suitable conditions.

MCMC sampling [24] approaches are based on performing randomly selected local moves in a state space $\mathcal{E}$ to approximately compute quantities of interest. The visited states $(X_0, X_1, \ldots)$ form a Markov chain, which under mild conditions converges to a stationary distribution $\pi$. Crucially, the probabilities of transitioning from one state to another are carefully chosen to ensure that the stationary distribution is identical to the distribution of interest. In our case, the state space is the powerset of $V$ (equivalently, the space of all binary vectors of length $n$), and to approximate the marginal probabilities of $p$ we construct a chain over subsets of $V$ that has stationary distribution $p$.

**The Gibbs sampler** In this paper, we focus on one of the simplest and most commonly used chains, namely the Gibbs sampler, also known as the Glauber chain. We denote by $P$ the transition matrix of the chain; each element $P(x, y)$ corresponds to the conditional probability of transitioning from state $x$ to state $y$, that is, $P(x, y) := \mathbb{P}[X_{t+1} = y \mid X_t = x]$, for any $x, y \in \mathcal{E}$, and any $t \geq 0$. We also define an adjacency relation $x \sim y$ on the elements of the state space, which denotes that $x$ and $y$ differ by exactly one element. It follows that each $x \in \mathcal{E}$ has exactly $n$ neighbors.

The Gibbs sampler is defined by an iterative two-step procedure, as shown in Algorithm 1. First, it selects an element $v \in V$ uniformly at random; then, it adds or removes $v$ to the current state $X_t$ according to the conditional probability of the resulting state. Importantly, the conditional probabilities that need to be computed do not depend on the partition function $Z$, thus the chain can be simulated efficiently, even though $Z$ is unknown and hard to compute. Moreover, it is easy to see that $\Delta_F(v|X_t) = \mathbb{1}_{\{v \notin X_t\}} F(v|X_t) + \mathbb{1}_{\{v \in X_t\}} F(v|X_t \setminus \{v\})$; thus, the sampler only requires a black box for the marginal gains of $F$, which are often faster to compute than the values of $F$ itself. Finally, it is easy to show that the stationary distribution of the chain constructed this way is $p$.

**Mixing times** Approximating quantities of interest using MCMC methods is based on using time averages to estimate expectations over the desired distribution. In particular, we estimate the expected value of function $f : \mathcal{E} \to \mathbb{R}$ by $\mathbb{E}_p[f(X)] \approx (1/T) \sum_{r=1}^{T} f(X_{s+r})$. For example, to estimate the marginal $p(v \in S)$, for some $v \in V$, we would define $f(x) = \mathbb{1}_{\{x_v = 1\}}$, for all $x \in \mathcal{E}$. The choice of burn-in time $s$ and number of samples $T$ in the above expression presents a tradeoff between computational efficiency and approximation accuracy. It turns out that the effect of both $s$ and $T$ is largely dependent on a fundamental quantity of the chain called *mixing time* [24].

The mixing time of a chain quantifies the number of iterations $t$ required for the distribution of $X_t$ to be close to the stationary distribution $\pi$. More formally, it is defined as $t_{\text{mix}}(\epsilon) := \min \{t \mid d(t) \leq \epsilon\}$, where $d(t)$ denotes the worst-case (over the starting state $X_0$ of the chain) total variation distance between the distribution of $X_t$ and $\pi$. Establishing upper bounds on the mix-

ing time of our Gibbs sampler is, therefore, sufficient to guarantee efficient approximate marginal inference (e.g., see [24, Theorem 12.19]).

## 4 Theoretical Results

In the previous section we mentioned that exact computation of the partition function for the class of models we consider here is, in general, infeasible. Only for very few exceptions, such as DPPs, is exact inference possible in polynomial time [23]. Even worse, it has been shown that the partition function of general Ising models is hard to approximate; in particular, there is no FPRAS for these models, unless RP = NP. [17] This implies that the mixing time of any Markov chain with such a stationary distribution will, in general, be exponential in $n$. It is, therefore, our aim to derive sufficient conditions that guarantee sub-exponential mixing times for the general class of models.

In some of our results we will use the fact that any submodular function $F$ can be written as

$$F = c + m + f, \tag{3}$$

where $c \in \mathbb{R}$ is a constant that has no effect on distributions defined by (1); $m$ is a normalized ($m(\varnothing) = 0$) modular function; and $f$ is a normalized ($f(\varnothing) = 0$) monotone submodular function, that is, it additionally satisfies the monotonicity property $f(v|S) \geq 0$, for all $v \in V$, and all $S \subseteq V$. A similar decomposition is possible for any supermodular function as well.

### 4.1 Polynomial-time mixing

Our guarantee for mixing times that are polynomial in $n$ depends crucially on the following quantity, which is defined for any set function $F : 2^V \to \mathbb{R}$:

$$\zeta_F := \max_{A,B \subseteq V} |F(A) + F(B) - F(A \cup B) - F(A \cap B)|.$$

Intuitively, $\zeta_F$ quantifies a notion of distance to modularity. To see this, note that a function $F$ is modular if and only if $F(A) + F(B) = F(A \cup B) + F(A \cap B)$, for all $A, B \subseteq V$. For modular functions, therefore, we have $\zeta_F = 0$. Furthermore, a function $F$ is submodular if and only if $F(A) + F(B) \geq F(A \cup B) + F(A \cap B)$, for all $A, B \subseteq V$. Similarly, $F$ is supermodular if the above holds with the sign reversed. It follows that for submodular and supermodular functions, $\zeta_F$ represents the worst-case amount by which $F$ violates the modular equality. It is also important to note that, for submodular and supermodular functions, $\zeta_F$ depends only on the monotone part of $F$; if we decompose $F$ according to (3), then it is easy to see that $\zeta_F = \zeta_f$. A trivial upper bound on $\zeta_F$, therefore, is $\zeta_F \leq f(V)$. Another quantity that has been used in the past to quantify the deviation of a submodular function from modularity is the curvature [4], defined as $\kappa_F := 1 - \min_{v \in V} (F(v|V \setminus \{v\})/F(v))$. Although of similar intuitive meaning, the multiplicative nature of its definition makes it significantly different from $\zeta_F$, which is defined additively.

As an example of a function class with $\zeta_F$ that do not depend on $n$, assume a ground set $V = \bigcup_{\ell=1}^{L} V_\ell$, and consider functions $F(S) = \sum_{\ell=1}^{L} \phi(|S \cap V_\ell|)$, where $\phi : \mathbb{R} \to \mathbb{R}$ is a bounded concave function, for example, $\phi(x) = \min\{\phi_{\max}, x\}$. Functions of this form are submodular, and have been used in applications such as document summarization to encourage diversity [25]. It is easy to see that, for such functions, $\zeta_F \leq L\phi_{\max}$, that is, $\zeta_F$ is independent of $n$.

The following theorem establishes a bound on the mixing time of the Gibbs sampler run on models of the form (1). The bound is exponential in $\zeta_F$, but polynomial in $n$.

**Theorem 1.** *For any function $F : 2^V \to \mathbb{R}$, the mixing time of the Gibbs sampler is bounded by*

$$t_{\mathrm{mix}}(\epsilon) \leq 2n^2 \exp(2\beta\zeta_F) \log\left(\frac{1}{\epsilon p_{\min}}\right),$$

*where $p_{\min} := \min_{S \in \mathcal{E}} p(S)$. If $F$ is submodular or supermodular, then the bound is improved to*

$$t_{\mathrm{mix}}(\epsilon) \leq 2n^2 \exp(\beta\zeta_f) \log\left(\frac{1}{\epsilon p_{\min}}\right).$$

Note that, since the factor of two that constitutes the difference between the two statements of the theorem lies in the exponent, it can have a significant impact on the above bounds. The dependence on $p_{\min}$ is related to the (worst-case) starting state of the chain, and can be eliminated if we have a way to guarantee a high-probability starting state. If $F$ is submodular or supermodular, this is usually straightforward to accomplish by using one of the standard constant-factor optimization algorithms [10, 26] as a preprocessing step. More generally, if $F$ is bounded by $0 \leq F(S) \leq F_{\max}$, for all $S \subseteq V$, then $\log(1/p_{\min}) = \mathcal{O}(n\beta F_{\max})$.

**Canonical paths**   Our proof of Theorem 1 is based on the method of *canonical paths* [5,15,16,28]. The high-level idea of this method is to view the state space as a graph, and try to construct a path between each pair of states that carries a certain amount of flow specified by the stationary distribution under consideration. Depending on the choice of these paths and the resulting load on the edges of the graph, we can derive bounds on the mixing time of the Markov chain.

More concretely, let us assume that for some set function $F$ and corresponding distribution $p$ as in (1), we construct the Gibbs chain on state space $\mathcal{E} = 2^V$ with transition matrix $P$. We can view the state space as a directed graph that has vertex set $\mathcal{E}$, and for any $A, B \in \mathcal{E}$, contains edge $(S, S')$ if and only if $S \sim S'$, that is, if and only if $S$ and $S'$ differ by exactly one element. Now, assume that, for any pair of states $A, B \in \mathcal{E}$, we define what is called a canonical path $\gamma_{AB} := (A = S_0, S_1, \ldots, S_\ell = B)$, such that all $(S_i, S_{i+1})$ are edges in the above graph. We denote the length of path $\gamma_{AB}$ by $|\gamma_{AB}|$, and define $Q(S, S') := p(S)P(S, S')$. We also denote the set of all pairs of states whose canonical path goes through $(S, S')$ by $\mathcal{C}_{SS'} := \{(A, B) \in \mathcal{E} \times \mathcal{E} \mid (S, S') \in \gamma_{AB}\}$. The following quantity, referred to as the *congestion* of an edge, uses a collection of canonical paths to quantify to what amount that edge is overloaded:

$$\rho(S, S') := \frac{1}{Q(S, S')} \sum_{(A,B) \in \mathcal{C}_{SS'}} p(A)p(B)|\gamma_{AB}|. \tag{4}$$

The denominator $Q(S, S')$ quantifies the capacity of edge $(S, S')$, while the sum represents the total flow through that edge according to the choice of canonical paths. The congestion of the whole graph is then defined as $\rho := \max_{S \sim S'} \rho(S, S')$. Low congestion implies that there are no bottlenecks in the state space, and the chain can move around fast, which also suggests rapid mixing. The following theorem makes this concrete.

**Theorem 2** ([15, 28]). *For any collection of canonical paths with congestion $\rho$, the mixing time of the chain is bounded by*

$$t_{\mathrm{mix}}(\epsilon) \leq \rho \log\left(\frac{1}{\epsilon p_{\min}}\right).$$

**Proof outline of Theorem 1**   To apply Theorem 2 to our class of distributions, we need to construct a set of canonical paths in the corresponding state space $2^V$, and upper bound the resulting congestion. First, note that, to transition from state $A \in \mathcal{E}$ to state $B \in \mathcal{E}$, in our case, it is enough to remove the elements of $A \setminus B$ and add the elements of $B \setminus A$. Each removal and addition corresponds to an edge in the state space graph, and the order of these operations identify a canonical path in this graph that connects $A$ to $B$. For our analysis, we assume a fixed order on $V$ (e.g., the natural order of the elements themselves), and perform the operations according to this order.

Having defined the set of canonical paths, we proceed to bounding the congestion $\rho(S, S')$ for any edge $(S, S')$. The main difficulty in bounding $\rho(S, S')$ is due to the sum in (4) over all pairs in $\mathcal{C}_{SS'}$. To simplify this sum we construct for each edge $(S, S')$ an injective map $\eta_{SS'} : \mathcal{C}_{SS'} \to \mathcal{E}$; this is a combinatorial encoding technique that has been previously used in similar proofs to ours [15]. We then prove the following key lemma about these maps.

**Lemma 1.** *For any $S \sim S'$, and any $A, B \in \mathcal{E}$, it holds that*

$$p(A)p(B) \leq 2n \exp(2\beta\zeta_F)Q(S, S')p(\eta_{SS'}(A, B)).$$

Since $\eta_{SS'}$ is injective, it follows that $\sum_{(A,B) \in \mathcal{C}_{SS'}} p(\eta_{SS'}(A, B)) \leq 1$. Furthermore, it is clear that each canonical path $\gamma_{AB}$ has length $|\gamma_{AB}| \leq n$, since we need to add and/or remove at most $n$ elements to get from state $A$ to state $B$. Combining these two facts with the above lemma, we get

$$\rho(S, S') \leq 2n^2 \exp(2\beta\zeta_F).$$

If $F$ is submodular or supermodular, we show that the dependence on $\zeta_F$ in Lemma 1 is improved to $\exp(\beta\zeta_F)$. More details can be found in the longer version of the paper.

## 4.2 Fast mixing

We now proceed to show that, under some stronger conditions, we are able to establish even faster—$\mathcal{O}(n \log n)$—mixing. For any function $F$, we denote $\Delta_F(v|S) := F(S \cup \{v\}) - F(S \setminus \{v\})$, and define the following quantity,

$$\gamma_{F,\beta} := \max_{\substack{S \subseteq V \\ r \in V}} \sum_{v \in V} \tanh\left(\frac{\beta}{2}\left|\Delta_F(v|S) - \Delta_F(v|S \cup \{r\})\right|\right),$$

which quantifies the (maximum) total influence of an element $r \in V$ on the values of $F$. For example, if the inclusion of $r$ makes no difference with respect to other elements of the ground set, we will have $\gamma_{F,\beta} = 0$. The following theorem establishes conditions for fast mixing of the Gibbs sampler when run on models of the form (1).

**Theorem 3.** *For any set function $F : 2^V \to \mathbb{R}$, if $\gamma_{F,\beta} < 1$, then the mixing time of the Gibbs sampler is bounded by*

$$t_{\mathrm{mix}}(\epsilon) \leq \frac{1}{1 - \gamma_{F,\beta}} n (\log n + \log \frac{1}{\epsilon}).$$

*If $F$ is additionally submodular or supermodular, and is decomposed according to* (3)*, then*

$$t_{\mathrm{mix}}(\epsilon) \leq \frac{1}{1 - \gamma_{f,\beta}} n (\log n + \log \frac{1}{\epsilon}).$$

Note that, in the second part of the theorem, $\gamma_{f,\beta}$ depends only on the monotone part of $F$. We have seen in Section 2 that some commonly used models are based on decomposable functions that can be written in the form (2). We prove the following corollary that provides an easy to check condition for fast mixing of the Gibbs sampler when $F$ is a decomposable submodular function.

**Corollary 1.** *For any submodular function $F$ that can be written in the form of* (2)*, with $f$ being its monotone (also decomposable) part according to* (3)*, if we define*

$$\theta_f := \max_{v \in V} \sum_{\ell \in [L]} \sqrt{f_\ell(v)} \quad and \quad \lambda_f := \max_{\ell \in [L]} \sum_{v \in V} \sqrt{f_\ell(v)},$$

*then it holds that*

$$\gamma_{f,\beta} \leq \frac{\beta}{2}\theta_f \lambda_f.$$

For example, applying this to the facility location model defined in Section 2, we get $\theta_f = \max_v \sum_{\ell=1}^L \sqrt{w_{v,\ell}}$, and $\lambda_f = \max_\ell \sum_{v \in V} \sqrt{w_{v,\ell}}$, and obtain fast mixing if $\theta_f \lambda_f \leq 2/\beta$. As a special case, if we consider the class of set cover functions ($w_{v,\ell} \in \{0, 1\}$), such that each $v \in V$ covers at most $\delta$ sets, and each set $\ell \in [L]$ is covered by at most $\delta$ elements, then $\theta_f, \lambda_f \leq \delta$, and we obtain fast mixing if $\delta^2 \leq 2/\beta$. Note, that the corollary can be trivially applied to any submodular function by taking $L = 1$, but may, in general, result in a loose bound if used that way.

**Coupling** Our proof of Theorem 3 is based on the *coupling* technique [1]; more specifically, we use the *path coupling* method [2,15,24]. Given a Markov chain $(X_t)$ on state space $\mathcal{E}$ with transition matrix $P$, a coupling for $(Z_t)$ is a new Markov chain $(X_t, Y_t)$ on state space $\mathcal{E} \times \mathcal{E}$, such that both $(X_t)$ and $(Y_t)$ are by themselves Markov chains with transition matrix $P$. The idea is to construct the coupling in such a way that, even when the starting points $X_0$ and $Y_0$ are different, the chains $(X_t)$ and $(Y_t)$ tend to coalesce. Then, it can be shown that the coupling time $t_{\mathrm{couple}} := \min\{t \geq 0 \mid X_t = Y_t\}$ is closely related to the mixing time of the original chain $(Z_t)$. [24]

The main difficulty in applying the coupling approach lies in the construction of the coupling itself, for which one needs to consider any possible pair of states $(Y_t, Z_t)$. The path coupling technique makes this construction easier by utilizing the same state-space graph that we used to define canonical paths in Section 4.1. The core idea is to first define a coupling only over adjacent states, and then extend it for any pair of states by using a metric on the graph. More concretely, let us denote by $d : \mathcal{E} \times \mathcal{E} \to \mathbb{R}$ the *path metric* on state space $\mathcal{E}$; that is, for any $x, y \in \mathcal{E}$, $d(x, y)$ is the minimum length of any path from $x$ to $y$ in the state space graph. The following theorem establishes fast mixing using this metric, as well as the diameter of the state space, $\mathrm{diam}(\mathcal{E}) := \max_{x,y \in \mathcal{E}} d(x, y)$.

**Theorem 4** ([2, 24]). *For any Markov chain $(Z_t)$, if $(X_t, Y_t)$ is a coupling, such that, for some $a \geq 0$, and any $x, y \in \mathcal{E}$ with $x \sim y$, it holds that*

$$\mathbb{E}[d(X_{t+1}, Y_{t+1}) \mid X_t = x, Y_t = y] \leq e^{-\alpha} d(x, y),$$

*then the mixing time of the original chain is bounded by*

$$t_{\mathrm{mix}}(\epsilon) \leq \frac{1}{\alpha} \left( \log(\mathrm{diam}(\mathcal{E})) + \log \frac{1}{\epsilon} \right).$$

**Proof outline of Theorem 3** In our case, the path metric $d$ is the Hamming distance between the binary vectors representing the states (equivalently, the number of elements by which two sets differ). We need to construct a suitable coupling $(X_t, Y_t)$ for any pair of states $x \sim y$. Consider the two corresponding sets $S, R \subseteq V$ that differ by exactly one element, and assume that $R = S \cup \{r\}$, for some $r \in V$. (The case $S = R \cup \{s\}$ for some $s \in V$ is completely analogous.) Remember that the Gibbs sampler first chooses an element $v \in V$ uniformly at random, and then adds or removes it according to the conditional probabilities. Our goal is to make the same updates happen to both $S$ and $R$ as frequently as possible. As a first step, we couple the candidate element for update $v \in V$ to always be the same in both chains. Then, we have to distinguish between the following cases.

If $v = r$, then the conditionals for both chains are identical, therefore we can couple both chains to add $r$ with probability $p_{\mathrm{add}} := p(S \cup \{r\})/(p(S) + p(S \cup \{r\}))$, which will result in new sets $S' = R' = S \cup \{r\}$, or remove $r$ with probability $1 - p_{\mathrm{add}}$, which will result in new sets $S' = R' = S$. Either way, we will have $d(S', R') = 0$.

If $v \neq r$, we cannot always couple the updates of the chains, because the conditional probabilities of the updates are different. In fact, we are forced to have different updates (one chain adding $v$, the other chain removing $v$) with probability equal to the difference of the corresponding conditionals, which we denote here by $p_{\mathrm{dif}}(v)$. If this is the case, we will have $d(S', R') = 2$, otherwise the chains will make the same update and will still differ only by element $r$, that is, $d(S', R') = 1$.

Putting together all the above, we get the following expected distance after one step:

$$\mathbb{E}[d(S', R')] = 1 - \frac{1}{n} + \frac{1}{n} \sum_{v \neq r} p_{\mathrm{dif}}(v) \leq 1 - \frac{1}{n}(1 - \gamma_{F,\beta}) \leq \exp\left(-\frac{1 - \gamma_{F,\beta}}{n}\right).$$

Our result follows from applying Theorem 4 with $\alpha = \gamma_{F,\beta}/n$, noting that $\mathrm{diam}(\mathcal{E}) = n$.

## 5 Experiments

We compare the Gibbs sampler against the variational approach proposed by Djolonga and Krause [6] for performing inference in models of the form (1), and use the same three models as in their experiments. We briefly review here the experimental setup and refer to their paper for more details.

The first is a (log-submodular) facility location model with an added modular term that penalizes the number of selected elements, that is, $p(S) \propto \exp(F(S) - 2|S|)$, where $F$ is a submodular facility location function. The model is constructed from randomly subsampling real data from a problem of sensor placement in a water distribution network [22]. In the experiments, we iteratively condition on random observations for each variable in the ground set. The second is a (log-supermodular) pairwise Markov random field (MRF; a generalized Ising model with varying weights), constructed by first randomly sampling points from a 2-D two-cluster Gaussian mixture model, and then introducing a pairwise potential for each pair of points with exponentially-decreasing weight in the distance of the pair. In the experiments, we iteratively condition on pairs of observations, one from each cluster. The third is a (log-supermodular) higher-order MRF, which is constructed by first generating a random Watts-Strogatz graph, and then creating one higher-order potential per node, which contains that node and all of its neighbors in the graph. The strength of the potentials is controlled by a parameter $\mu$, which is closely related to the curvature of the functions that define them. In the experiments, we vary this parameter from 0 (modular model) to 1 ("strongly" supermodular model).

For all three models, we constrain the size of the ground set to $n = 20$, so that we are able to compute, and compare against, the exact marginals. Furthermore, we run multiple repetitions for each model to account for the randomness of the model instance, and the random initialization of

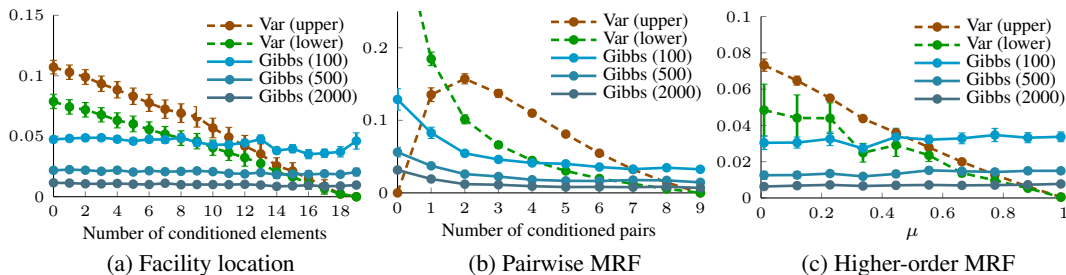

Figure 1: Absolute error of the marginals computed by the Gibbs sampler compared to variational inference [6]. A modest 500 Gibbs iterations outperform the variational method for the most part.

the Gibbs sampler. The marginals we compute are of the form $p(v \in S \mid C \subseteq S \subseteq D)$, for all $v \in V$. We run the Gibbs sampler for 100, 500, and 2000 iterations on each problem instance. In compliance with recommended MCMC practice [11], we discard the first half of the obtained samples as burn-in, and only use the second half for estimating the marginals.

Figure 1 compares the average absolute error of the approximate marginals with respect to the exact ones. The averaging is performed over $v \in V$, and over the different repetitions of each experiment; errorbars depict two standard errors. The two variational approximations are obtained from factorized distributions associated with modular lower and upper bounds respectively [6]. We notice a similar trend on all three models. For the regimes that correspond to less "peaked" posterior distributions (small number of conditioned variables, small $\mu$), even 100 Gibbs iterations outperform both variational approximations. The latter gain an advantage when the posterior is concentrated around only a few states, which happens after having conditioned on almost all variables in the first two models, or for $\mu$ close to 1 in the third model.

## 6 Further Related Work

In contemporary work to ours, Rebeschini and Karbasi [27] analyzed the mixing times of log-submodular models. Using a method based on matrix norms, which was previously introduced by Dyer et al. [7], and is closely related to path coupling, they arrive at a similar—though not directly comparable—condition to the one we presented in Theorem 3.

Iyer and Bilmes [13] recently considered a different class of probabilistic models, called submodular point processes, which are also defined through submodular functions, and have the form $p(S) \propto F(S)$. They showed that inference in SPPs is, in general, also a hard problem, and provided approximations and closed-form solutions for some subclasses.

The canonical path method for bounding mixing times has been previously used in applications, such as approximating the partition function of ferromagnetic Ising models [17], approximating matrix permanents [16, 18], and counting matchings in graphs [15]. The most prominent application of coupling-based methods is counting $k$-colorings in low-degree graphs [3, 14, 15]. Other applications include counting independent sets in graphs [8], and approximating the partition function of various subclasses of Ising models at high temperatures [24].

## 7 Conclusion

We considered the problem of performing marginal inference using MCMC sampling techniques in probabilistic models defined through submodular functions. In particular, we presented for the first time sufficient conditions to obtain upper bounds on the mixing time of the Gibbs sampler in general log-submodular and log-supermodular models. Furthermore, we demonstrated that, in practice, the Gibbs sampler compares favorably to previously proposed variational approximations, at least in regimes of high uncertainty. We believe that this is an important step towards a unified framework for further analysis and practical application of this rich class of probabilistic submodular models.

**Acknowledgments**  This work was partially supported by ERC Starting Grant 307036.

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
