[Supplementary Material]

# A  Polynomial-time Mixing Proofs

For sets $A, B$, we denote $A \oplus B := (A \setminus B) \cup (B \setminus A)$.

**Lemma A.1.** *Define the maps $\eta_{SS'} : \mathcal{C}_{SS'} \to \mathcal{E}$, for each pair $(S, S') \in \mathcal{E} \times \mathcal{E}$ with $S \sim S'$, as follows:*

$$\eta_{SS'}(A, B) = \begin{cases} A \oplus B \oplus S, & \text{if } F(S') \geq F(S) \\ A \oplus B \oplus S', & \text{otherwise} \end{cases}.$$

*Then, each map $\eta_{SS'}$ is injective.*

*Proof.* Assume that $F(S') \geq F(S)$, and $S' = S \cup \{r\}$, for some $r \in V$. Assume that we are given $C := A \oplus B \oplus S$, and we want to recover $A$ and $B$. We will denote by $\prec$ the natural ordering of the ground set $V$. First, we define

$$K^- := \{v \in C \oplus S \mid v \prec r\}$$
$$K^+ := \{v \in C \oplus S \mid v \succ r\}.$$

Then, we can recover $A$ and $V$ as follows:

$$A = S \oplus K^-$$
$$B = S' \oplus K^+.$$

The case $S' = S \setminus \{r\}$, as well as the two cases for $F(S') < F(S)$ are completely analogous. Note that the distinction based on the value of the function has no effect on the proof here, but is technically needed for the next lemma. The only thing that changes between the cases is whether the element $r$ that gets added or removed in the transition $(S, S')$ belongs to $A$ or $B$, which is always straightforward to determine from the type of the transition (for additions it belongs to $B$, and for removals to $A$). $\square$

**Lemma 1.** *For any $S \sim S'$, and any $A, B \in \mathcal{E}$, it holds that*

$$p(A)p(B) \leq 2n \exp(2\beta\zeta_F) Q(S, S') p(\eta_{SS'}(A, B)).$$

*If $F$ is submodular or supermodular, then the bound is improved to*

$$p(A)p(B) \leq 2n \exp(\beta\zeta_f) Q(S, S') p(\eta_{SS'}(A, B)).$$

*Proof.* We will consider the case $S' = S \cup \{r\}$, for some $r \in V$, with $F(S') \geq F(S)$. Again, the other three cases are completely analogous by using $\eta_{SS'}$ as defined in Lemma A.1.

We first compute

$$
\begin{aligned}
Q(S, S') &= p(S)P(S, S') \\
&= \frac{1}{n} \frac{p(S)p(S')}{p(S) + p(S')} && \text{by definition of the Gibbs sampler} \\
&= \frac{1}{nZ} \frac{\exp(\beta F(S)) \exp(\beta F(S'))}{\exp(\beta F(S)) + \exp(\beta F(S'))} && \text{by definition of our models} \\
&\geq \frac{1}{nZ} \frac{\exp(\beta F(S)) \exp(\beta F(S'))}{2 \exp(\beta F(S'))} && \text{by } F(S') \geq F(S) \\
&= \frac{\exp(\beta F(S))}{2nZ}.
\end{aligned}
$$

As a result, we get

$$\frac{p(A)p(B)}{Q(S, S')} \leq \frac{2n}{Z} \exp(\beta(F(A) + F(B) - F(S))). \tag{5}$$

Let us denote $\zeta_F(A, B) := F(A) + F(B) - F(A \cup B) - F(A \cap B)$, for any $A, B \subseteq V$, so that $\zeta_F = \max_{A,B \subseteq V} |\zeta_F(A, B)|$. Then, if we denote $C := \eta_{SS'}(A, B) = A \oplus B \oplus S$, we have

$F(A) + F(B) - F(S)$
$= (F(A) + F(B) - F(A \cup B) - F(A \cap B)) - (F(S) + F(C) - F(A \cup B) - F(A \cap B)) + F(C)$
$= (F(A) + F(B) - F(A \cup B) - F(A \cap B)) - (F(S) + F(C) - F(S \cup C) - F(S \cap C)) + F(C)$
$= \zeta_F(A, B) - \zeta_F(S, C) + F(C)$
$\leq 2\zeta_F + F(C).$

If $F$ is submodular, then $\zeta_F(A, B)$ and $\zeta_F(S, C)$ are both non-negative, therefore $\zeta_F(A, B) - \zeta_F(S, C) + F(C) \leq \zeta_F + F(C) = \zeta_f + F(C)$. Similarly, if $F$ is supermodular, then $\zeta_F(A, B)$ and $\zeta_F(S, C)$ are both non-positive, therefore $\zeta_F(A, B) - \zeta_F(S, C) + F(C) \leq \zeta_F + F(C) = \zeta_f + F(C)$. Substituting these bounds in (5) gives us the result of the lemma. $\square$

# B  Fast Mixing Proofs

**Lemma B.1.** *For any $S, R \subseteq V$ with $R = S \cup \{r\}$, if we define*

$$p_{\mathrm{dif}}(v) := \left| \frac{p(S \cup \{v\})}{p(S \cup \{v\}) + p(S \setminus \{v\})} - \frac{p(R \cup \{v\})}{p(R \cup \{v\}) + p(R \setminus \{v\})} \right|,$$

*then it holds that*

$$\sum_{v \neq r} p_{\mathrm{dif}}(v) \leq \gamma_{F,\beta}.$$

*Proof.* For any $v \neq r$, we have

$$p_{\mathrm{dif}}(v) = \left| \frac{\exp(\beta F(S \cup \{v\}))}{\exp(\beta F(S \cup \{v\})) + \exp(\beta F(S \setminus \{v\}))} - \frac{\exp(\beta F(R \cup \{v\}))}{\exp(\beta F(R \cup \{v\})) + \exp(\beta F(R \setminus \{v\}))} \right|$$

$$= \left| \frac{\exp(\beta \Delta_F(v|S))}{1 + \exp(\beta \Delta_F(v|S))} - \frac{\exp(\beta \Delta_F(v|R))}{1 + \exp(\beta \Delta_F(v|R))} \right|$$

$$= \left| \frac{\exp(\beta \Delta_F(v|S)) - \exp(\beta \Delta_F(v|R))}{(1 + \exp(\beta \Delta_F(v|S)))(1 + \exp(\beta \Delta_F(v|R)))} \right|$$

$$\leq \left| \frac{\exp(\beta \Delta_F(v|S)) - \exp(\beta \Delta_F(v|R))}{\exp(\beta \Delta_F(v|S)) + \exp(\beta \Delta_F(v|R))} \right|$$

$$= \left| \frac{\exp(\beta (\Delta_F(v|S) - \Delta_F(v|R))) - 1}{\exp(\beta (\Delta_F(v|S) - \Delta_F(v|R))) + 1} \right|$$

$$= \tanh\left( \frac{\beta}{2} |(\Delta_F(v|S) - \Delta_F(v|R))| \right).$$

The lemma follows then by the definition of $\gamma_{F,\beta}$, and the fact that $R = S \cup \{r\}$. $\square$

**Lemma B.2.** *If $F$ is submodular or supermodular, and decomposed according to* (3)*, then*

$$\gamma_{F,\beta} = \gamma_{f,\beta}.$$

*Proof.* For any $S, R \subseteq V$ with $R = S \cup \{r\}$, and any $v \in V$, we have

$$\Delta_F(v|S) - \Delta_F(v|R) = F(S \cup \{v\}) - F(S \setminus \{v\}) - F(R \cup \{v\}) + F(R \setminus \{v\})$$
$$= f(S \cup \{v\}) - f(S \setminus \{v\}) - f(R \cup \{v\}) + f(R \setminus \{v\})$$
$$= \Delta_f(v|S) - \Delta_f(v|R).$$

$\square$

**Corollary 2.** *For any submodular function $F$ that can be written in the form of* (2)*, with $f$ being its monotone (also decomposable) part according to* (3)*, if we define*

$$\theta_f := \max_{v \in V} \sum_{\ell \in [L]} \sqrt{f_\ell(v)} \quad and \quad \lambda_f := \max_{\ell \in [L]} \sum_{v \in V} \sqrt{f_\ell(v)},$$

*then it holds that*

$$\gamma_{f,\beta} \le \frac{\beta}{2} \theta_f \lambda_f.$$

*Proof.* For any $S, R \subseteq V$ with $R = S \cup \{r\}$, we have

$$\sum_{v \ne r} \tanh\left( \frac{\beta}{2} \left| (\Delta_f(v|S) - \Delta_f(v|R)) \right| \right)$$

$$\le \sum_{v \ne r} \frac{\beta}{2} \left| (\Delta_f(v|S) - \Delta_f(v|R)) \right| \qquad\qquad \text{by } \tanh(x) \le x, \text{ for all } x \ge 0$$

$$\le \sum_{v \ne r} \frac{\beta}{2} (\Delta_f(v|S) - \Delta_f(v|R)) \qquad\qquad \text{by submodularity of } f$$

$$= \frac{\beta}{2} \sum_{v \ne r} (f(S \cup \{v\}) - f(S \setminus \{v\}) - f(S \cup \{r\} \cup \{v\}) + f(S \cup \{r\} \setminus \{v\}))$$

$$= \frac{\beta}{2} \sum_{v \ne r} \sum_{\ell \in [L]} (f_\ell(S \cup \{v\}) - f_\ell(S \setminus \{v\}) - f_\ell(S \cup \{r\} \cup \{v\}) + f_\ell(S \cup \{r\} \setminus \{v\}))$$

$$\le \frac{\beta}{2} \sum_{v \ne r} \sum_{\ell \in [L]} \min \left\{ f_\ell(S \cup \{v\}) - f_\ell(S \setminus \{v\}), f_\ell(S \cup \{r\} \setminus \{v\}) - f_\ell(S \setminus \{v\}) \right\}$$

$$\qquad\qquad\qquad\qquad\qquad\qquad\qquad\qquad\qquad\qquad\qquad \text{by monotonicity of } f_\ell$$

$$\le \frac{\beta}{2} \sum_{v \ne r} \sum_{\ell \in [L]} \min \left\{ f_\ell(v), f_\ell(r) \right\} \qquad\qquad \text{by submodularity of } f_\ell$$

$$\le \frac{\beta}{2} \sum_{v \ne r} \sum_{\ell \in [L]} \sqrt{f_\ell(v) f_\ell(r)}$$

$$= \frac{\beta}{2} \sum_{\ell \in [L]} \sqrt{f_\ell(r)} \sum_{v \ne r} \sqrt{f_\ell(v)}.$$

The result follows by maximizing both sides over $S$ and $r$. $\qquad\qquad\qquad\qquad\qquad\qquad$ $\square$