[Reviews · NeurIPS 2015]

Submitted by Assigned_Reviewer_1

Summary of the paper The paper investigates sampling from submodular or supermodular distributions, that is, distributions over the power set of a finite set, which can be described as the exponential of a submodular or supermodular function. Sufficient conditions are given such that a vanilla Glauber MCMC algorithm has a fast mixing time.

Summary of the review The paper is very well written. The topic is potentially impactful, as submodular distributions are good candidates for large-scale distributions that could potentially be efficiently sampled. The results on the mixing times are interesting. My main concern is that the link between the theoretical results of Section 4 and the experiments of Section 5 is not obvious, as the experiments do not consider the scaling

in the size of the ground set.

Major comments L373 How do the results of Section 4 apply to the two MRFs considered here? How about determinantal point processes mentioned in the introduction? L394 It'd be interesting to have an experiment across different values of n to have a rough idea of how the practical speed of convergence relates to the mixing times of Section 4.

L407 Can you further explain why the variational approximations gain an advantage?

Summary: The paper is very well written. The topic is potentially impactful, as submodular distributions are good candidates for large-scale distributions that could potentially be efficiently sampled. The results on the mixing times are interesting. My main concern is that the link between the theoretical results of Section 4 and the experiments of Section 5 is not obvious, as the experiments do not consider the scaling

in the size of the ground set.

Submitted by Assigned_Reviewer_2

[This is a "light" review.]

This paper proves bounds for the mixing time of a Gibbs sampler that performs random single-site include/exclude updates on sets with log-submodular or log-supermodular probability mass functions.

In the case where there are many possible "low congestion" paths between states, this mixing is shown to have polynomial time.

This is an exciting result, although I have not verified the proof.

The thing that confuses me, and that I have to assume is simply a typo that does not implicate the main result is that Algorithm 1 cannot be a correct Gibbs sampler for the target distribution.

The probability of adding v into the set appears to be proportional to \Delta F / (1 + \Delta F) but \Delta F is the change in the _log_ of the probability mass function.

I would expect the \Delta F to have exponential functions around them, as well as \beta.

In fact, one way you can be certain that Algorithm 1 is incorrect is that the inverse temperature doesn't appear anywhere; at very low temperatures (large \beta) it should want to stick in the mode and at very high temperatures (small \beta) it should be essentially uniform.

I feel that probably this is just a typo and does not impugn the main result because the definition of \gamma_{F,\beta} in Section 4.2 looks sensible and small \beta would push down the mixing time, as expected.
Summary: This is a nice paper with an interesting and useful result, assuming that Algorithm 1 just has typos relative to the proofs.

Submitted by Assigned_Reviewer_3

The paper addresses an interesting problem and provides interesting results. Submodular models have become popular recently in machine learning and this is the first paper I am aware of analyzing sampling procedures in this context.

This is a very solid paper which should be published in a top venue.

Quality: very good

Clarity: very good

Originality: new to the best of my knowledge

Significance: I am really not sure NIPS is an appropriate venue for this paper, hence my mark.
Summary: This paper analyzes the performance of the Gibbs sampler for probabilistic submodular models. It presents interesting theoretical results based on the method of canonical paths.

Submitted by Assigned_Reviewer_4

Summary: The paper investigates sampling from discrete distributions defined in terms of a sub- or supermodular energy function. In particular, the mixing of the single-site Gibbs sampler is analyzed. Conditions are given guaranteeing O(n^2) and O(n log n) mixing.

The paper is generally clear and well-written. The theoretical results regarding mixing seem solid, though I have not checked the proofs carefully. I do wonder at the usefulness of the class of probabilistic sub/supermodular models. The authors claim it is a "rich" model class, but the examples of such models that are given are Ising models, DPPs, and facility location models. The first two are already well-understood and it's not clear why the facility location model needs to be probabilistic. The models used in the experiments all seem quite contrived, though the experiments do a good job of showing the strengths and weaknesses of the Gibbs sampler compared to an existing variational approach.

Other comments: l21: models, => models l34: "similarly" is a poor word choice l42-50: grammar issues l113: should be F(v|X_t \cup \{v\}) - F(v | X_T \setminus \{v\}) l151: examples of when the marginal gain is faster to compute?
Summary: A solid paper that analyzes the single-site Gibbs sampler for probabilistic sub/supermodular models. The extent to which this model class is useful is unclear, however.

Submitted by Assigned_Reviewer_5

The authors provide a formal analysis of the mixing time of simple Gibbs sampling schemes on probabilistic submodular models. The main result is that the mixing time of Gibbs is polynomial or even less nlog(n) with the dimensionality on submodular models. To my knowledge, this work is novel and the result is interesting.

Quality: The writing of this paper is clear and most of the paper is easy to follow. However, the author gives little introductions on some key results, like Therom 2, so it is not well self-contained. This makes it hard to understand for people not familiar the specific literature.

Strength: The bound of mixing time is important for MCMC methods, but often difficult to establish. This work provides a novel result of the mixing time using submodularity.

Questions:

In addition, the proof of two main theorems on the bounds is not clear to me. More specific: Theorem 1: The idea of the set of canonical path is clear, but the definition C_{s,s'} seems simply the product space of two configurations, rather than a canonical path between them. This makes it difficult to understand the injective map eta_{s,s'}.

Theorem 3: It is not clear to me what is the relationship between the chain (X_t, Y_t) and set (R, S). Is (R,S) the current sample from the pair of chains? In particular, what does it mean the two chain are identical if v = r?

One more point is that it is good have more discussion on how realistic the strong condition for fast mixing. It seems very strong to have gamma_{f, beta} < 1 on most interesting models.

I think the author should be careful about not over advertising the polynomial mixing time, because the bound is actually exponential to zeta, the notation of distance to modular function. It can be very large in many models, so the bounds may not be a good guid of real mixing time in practice.

For the experiment, it is good to compare Gibbs with variational method, but it is not clear to me to see the bound of mixing time in the result.
Summary: This is an interesting theoretical paper aiming at the bound of mixing time of Gibbs sampling on submodular models, but some important points in the proof is lack of clarity. More experiments are suggested to support the analytical result.

Author Feedback
Author rebuttal: We would like to thank all reviewers for their helpful feedback.

Assigned_Reviewer_1
-------------------
We agree that investigating empirical convergence behavior as a function of problem size is an interesting direction for future work. This is, however, a difficult undertaking, since (1) selecting a suitable class of instances that can naturally be scaled in terms of size (n) is not always straightforward, and (2) diagnosing convergence of a MCMC algorithm in practice is known to be notoriously difficult. That is why, in this work, we elected to evaluate, and compare with the variational baseline, the accuracy of the sampler on previously-proposed practical instances.

Assigned_Reviewer_2
-------------------
* Similarly to other probabilistic models, having a probabilistic facility location model allows us to quantify uncertainty about conclusions we draw using the model, and provides a principled way for learning such a model from data, with applications including document [1] and image [2] summarization.

* As a particular example of more complex models that fall into the log-sub-/supermodular class, consider Ising-style models that additionally contain higher-order potentials. Such models have been recently introduced to allow for incorporating superpixel information into image segmentation problems [3].

* For a graph-cut function (used in the Ising model), computing a marginal gain F(v|S) requires O(degree(v)) computations, whereas computing the value of the function F(S) requires O(number-of-cut-edges(S)) computations.

[1] Hui Lin and Jeff Bilmes. A class of submodular functions for document summarization. ACL, 2011.
[2] Sebastian Tschiatschek, Rishabh Iyer, Haochen Wei, and Jeff Bilmes. Learning mixtures of submodular functions for image collection summarization. NIPS, 2014.
[3] Josip Djolonga and Andreas Krause. Scalable variational inference in log-supermodular models. ICML, 2015.

Assigned_Reviewer_3
-------------------
* C_{S, S'} is indeed a subset of the product space of states, which means that it also enumerates the canonical paths that go through edge (S, S'). (Each ordered pair of states has an associated canonical path.) It is hard to give an intuition about the injective map eta_{S, S'}, but it is technically a way to get an upper bound on the sum of equation (4), as shown in Lemma 1.

* Yes, S and R are the assumed current states of the two chains X_t and Y_t. (In fact, the current states are x and y, represented as binary vectors, while S and R are the corresponding sets.) When v = r, it means that the Gibbs sampler has chosen to update (add or remove) element r, which is exactly the difference between S and R. This means that the probability of adding r to S is equal to the probability of keeping r in R. Similarly, the probability of not adding r to S is equal to the probability of removing r from R. This is why, in this case, we can couple the two chains "without losing anything".

* Regarding our experiments, please see our response to Assigned_Reviewer_1.

Assigned_Reviewer_4
-------------------
We believe that efficient inference in probabilistic models like the ones we consider is of interest to the NIPS community, as shown by a number of recently published papers related to this topic (e.g., [4], [5]).

[4] Josip Djolonga and Andreas Krause. From MAP to marginals: Variational inference in bayesian submodular models. NIPS, 2014.
[5] Xianghang Liu and Justin Domke. Projecting markov random field parameters for fast mixing. NIPS, 2014.

Assigned_Reviewer_5
-------------------
It is indeed a typo; thanks for pointing it out! Lines 4 and 5 of the algorithm should be as follows:

4: \Delta_F(v | X_t) <- F(X_t U {v}) - F(X_t \ {v})
5: p_add <- exp(\beta \Delta_F(v | X_t)) / (1 + exp(\beta \Delta_F(v | X_t)))

The definition of \Delta_F in line 277 should also be changed to reflect this fix:
\Delta_F(v | S) := F(S U {v}) - F(S \ {v})
= I(v \not\in S) F(v | S) + I(v \in S) F(v | S \ {v}) [here I(.) denotes the indicator function]